# Secure Bluetooth Communication in Smart Healthcare Systems: A Novel Community Dataset and Intrusion Detection System [note 1]

**DOI:** 10.3390/s22218280

**Published:** 2022-10-28

**Authors:** Mohammed Zubair, Ali Ghubaish, Devrim Unal, Abdulla Al-Ali, Thomas Reimann, Guillaume Alinier, Mohammad Hammoudeh, Junaid Qadir

**Affiliations:** 1Kindi Center for Computing Research, Qatar University, Doha P.O. Box 2713, Qatar; 2Department of Computer Science, Qatar University, Doha P.O. Box 2713, Qatar; 3Department of Computer Science and Engineering, Washington University in St. Louis, St. Louis, MO 63130, USA; 4Copenhagen Emergency Medical Service, 3400 Hillerød, Denmark; 5Department of Emergency Management, Jacksonville State University, Alabama, AL 36265, USA; 6Hamad Medical Corporation Ambulance Service, Doha P.O. Box 3050, Qatar; 7School of Health and Social Work, University of Hertfordshire, Hatfield AL10 9AB, UK; 8Weil Cornell Medicine, Doha P.O. Box 24144, Qatar; 9Faculty of Health and Life Sciences, Northumbria University, Newcastle upon Tyne NE1 8ST, UK; 10Information and Computer Science Department, King Fahd University of Petroleum and Minerals, Dhahran 31261, Saudi Arabia

**Keywords:** smart city networks, wireless communications, Bluetooth, artificial intelligence, communication security

## Abstract

Smart health presents an ever-expanding attack surface due to the continuous adoption of a broad variety of Internet of Medical Things (IoMT) devices and applications. IoMT is a common approach to smart city solutions that deliver long-term benefits to critical infrastructures, such as smart healthcare. Many of the IoMT devices in smart cities use Bluetooth technology for short-range communication due to its flexibility, low resource consumption, and flexibility. As smart healthcare applications rely on distributed control optimization, artificial intelligence (AI) and deep learning (DL) offer effective approaches to mitigate cyber-attacks. This paper presents a decentralized, predictive, DL-based process to autonomously detect and block malicious traffic and provide an end-to-end defense against network attacks in IoMT devices. Furthermore, we provide the *BlueTack* dataset for Bluetooth-based attacks against IoMT networks. To the best of our knowledge, this is the first intrusion detection dataset for Bluetooth classic and Bluetooth low energy (BLE). Using the BlueTack dataset, we devised a multi-layer intrusion detection method that uses deep-learning techniques. We propose a decentralized architecture for deploying this intrusion detection system on the edge nodes of a smart healthcare system that may be deployed in a smart city. The presented multi-layer intrusion detection models achieve performances in the range of 97–99.5% based on the F1 scores.

## 1. Introduction

Cities are being transformed into *smart cities* via Internet-of-Things (IoT) technology. Smart cities use technologies for sensing, networking, and computation to enhance the quality of life and well-being of inhabitants. Such smart cities also require new service-centric computing paradigms for next-generation networks (5G, 6G, and beyond) [1]. While there are numerous networking technologies available for long-range communications, the most widely used technology for close-proximity communications is Bluetooth. Bluetooth is well suited for operations on resource-constrained mobile devices due to its low power consumption, low cost, and support for multimedia, such as data and audio streaming. Bluetooth is also widely used in smart healthcare systems to enable untethered wireless communications between smart healthcare devices. Recently, Bluetooth was prominent in its adoption for contact-tracing applications in the fight against the COVID-19 global pandemic [2].

By the year 2030 [3], the number of IoT devices is expected to surge by 124 billion. Moreover, the healthcare economy statistics predict that the market for IoT devices will grow from USD 20 billion in 2015 to USD 70 billion in 2025. It was also reported that 30.3% of the IoT devices in use are in the health sector [4]. The massive deployment of IoT devices in heterogeneous networks with multiple technologies and protocols (such as Wi-Fi, long-term evolution (LTE), Bluetooth, and ZigBee) makes the task of securing such networks very complex. Research from the Information Systems Audit and Control Association (ISACA) [5] on smart cities identified the security of IoT devices as important, as numerous smart city critical infrastructure (CI) concepts (e.g., intelligent transport, healthcare system, and energy distribution) rely on the robustness and security of smart technologies and IoT devices [6].

As the number of Internet of Medical Things (IoMT) devices increases, the network becomes congested, which leads to bandwidth and latency bottlenecks [7]. For instance, an IoMT device sends data to a medical professional for regular analysis. This transmission of data to the cloud can potentially cause latency and bandwidth congestion in the communication path [8], which could endanger the life of the patient. To address this challenge, the edge cloud concept has emerged for the IoMT paradigm. An edge cloud improves efficiency and provides more reliability for the smart healthcare system. The quick response time and reduced energy consumption will result in longer battery life for medical devices and reduce the usage of network bandwidth [9,10].

The exponential growth of IoT devices and the massive interconnectivity between such devices greatly opens up the potential attack surface for smart healthcare services that may be exploited by malicious actors. IoT devices are vulnerable to various medium- and high-severity attacks [11]. Various vulnerabilities allow the intruders to perform a wide range of attacks, such as denial of service (DoS), distributed DoS (DDoS), man-in-the-middle (MITM), data leakage, and spoofing. These attacks result in the unavailability of system resources and can lead to physical harm to the individuals when the patient is ambulance-bound or hospital-bound. According to a report from the Global Connected Industries Cybersecurity, 82% of healthcare facilities experience cyber-attacks, amongst which, 30% target IoT devices [11]. The potential weakness in the network, IoT device, and protocol allows the attackers to access the network completely in an unauthorized way (e.g., Mirai attack) [12]. Apart from these cyber-attacks, insecure operating systems, and application vulnerabilities are other major threats to the healthcare system. Investigations show that 83% of IoT devices run on outdated operating systems, and around 51% of the cyber threats in the health sector concern imaging devices, which lead to the disruption of communication between patients and medical professionals. Moreover, 98% of IoT device traffic is in plain text that can be intercepted by adversaries.

Traditional security mechanisms cannot be enforced in the IoT network because the network protocol stack itself may have numerous vulnerabilities. Zero-day attacks are very difficult to be detected by traditional security mechanisms due to computational expenses, which do not go well with the resource-constrained nature of typical IoT devices [13]. Conventional perimeter security controls only defend against external attacks, but they fail to detect internal attacks within the network. An intelligent and faster detection mechanism is required to guarantee the security of the IoT network for countering new threats before the network is compromised.

In this paper, our focus is on the security of Bluetooth communication in smart healthcare systems. After reviewing the significant security problems, we focus on the detection of wireless attacks against IoMT. Wireless attacks are performed when the data are at rest or in transmission from one device to another device in a wireless medium over different channels using various protocols, namely Bluetooth low energy (BLE), Bluetooth basic rate/ enhanced data rate (BR/EDR), Wi-Fi, long-range (LoRA), etc. The openness of the wireless network poses threats to the entire network and can end up compromising the entire system. The attacker may perform various attacks, such as peer-to-peer, denial-of-service, eavesdropping, man-in-the-middle (MITM), and authentication attacks to take over the IoMT device or complete network. The main contributions of this study are as follows:We curated a novel first-of-its-kind *BlueTack dataset* for Bluetooth-based IoT attacks. The BlueTack dataset consists of popular attacks against Bluetooth BR/EDR or Bluetooth classic protocols, namely: Bluesmack, DoS, DDoS, and similar attacks, such as DDoS and MITM attacks on the BLE protocol. To the best of our knowledge, this is the first intrusion detection dataset for the Bluetooth classic protocol and BLE. The BlueTack dataset will be made publicly accessible as described in the Dataset Availability Statement.A secure and scalable framework for the deployment of an intrusion detection system(s) (IDS) on the edge nodes of IoT-based healthcare systems in smart cities. The framework guarantees quicker identification of malicious activities to ensure the safety of critically ill patients transported by ambulances.A multi-layer intrusion detection model using deep learning (DL) to protect the edge nodes of the smart healthcare IoMT system. Since IoMT is composed of several resource-constrained devices, deploying the DL model on the IoMT device itself for advanced functionality is impractical. Hence, The IDS is divided into two layers: *Layer*_1 (where preprocessing is performed on IoMT devices or the edge node) and *Layer*_2 (a standalone GPU capability device in which the DL model is deployed). The proposed DL-based IDS achieves 99% accuracy while being deployed in a real-time scenario.

The flow of this paper is structured as follows: Section 2 provides an overview of related work, followed by Section 3, which illustrates the proposed model, architecture, and dataset in detail. We show the results of the performance evaluation of the proposed model in Section 4. Finally, the work is concluded with future directions in Section 5.

## 2. Related Work

Before we introduce the methodology, we discuss the background and related work available in the literature.

### 2.1. Security of IoMT

IoMT devices perform diverse tasks in smart healthcare systems, such as recording electrical impulses through electrocardiograms (ECGs) or monitoring blood glucose or blood pressure. For ambulance-bound patients, IoMT devices monitor the patient’s activity, save critical information about the patient’s physiological signals, and trigger alerts to the medical staff inside the ambulance or a remote monitoring device through the cloud. As the complete information of the patient flows in and out through the IoMT gateway [14], securing the IoMT attack surface assumes critical importance. An attacker may target the IoMT gateway to manipulate information before sending it to the doctor or to launch denial of service attacks to make the information unavailable. Such malevolent activities can put the patient’s life at risk. Rasool et al. [15] reviewed various security issues of IoMT devices. The authors describe the vulnerabilities that exist in these devices, which can be exploited by attackers easily. In our article, we consider internal and external threats that are targeted against IoMT infrastructure. Since these devices are severely resource-constrained, it is easy to render these devices unavailable by draining their battery with devastating implications [16]. Thus, our focus in this paper is on attacks that may drain the batteries of these devices or that make the devices unavailable due to multiple ping requests.

### 2.2. Communication in Smart Healthcare System

The typical architecture of a smart healthcare system is shown in Figure 1. A typical smart healthcare system comprises three domains: IoT domain, cloud domain, and user domain, which generate data, store data, and make diagnoses, respectively. The *IoT domain* consists of wireless medical devices, actuators, sensors, gateways, and other devices. Here, the focus is on acquiring patients’ data from IoMT devices and transmitting it to the cloud for storage and subsequent access. The *cloud domain* is stratified by the edge and core cloud. The edge cloud is placed on the premises of the medical facility to ensure continuous connectivity and low latency, in addition to quicker diagnosis of acute cases. The core cloud provides massive storage and comprehensive analysis of data, and it helps in the diagnosis of current symptoms based on previous related records.

During IoMT communication, the vital information of a patient is maintained by an electronic patient care record device (EPCRD), which is commonly known as a Toughpad. It has the capability of integrating different communication protocols and it acts as a gateway for Bluetooth, Wi-Fi, and long-term evolution (LTE) communication. Furthermore, the EPCRD acts as an edge device that allows and enables the technologies for computation at the edge of the healthcare network. It accomplishes the tasks of caching, processing storage, computation offloading, request distribution, and delivery of the services from the cloud end to the user end. In our proposed approach, we leverage edge cloud technology and deploy the IDS on the edge nodes of the healthcare system. The *user domain* delivers the processed data from other domains to the authorized clinical staff. Integration and streaming of vast volumes of data from different sources are visualized in various forms, such as graphics, images, tabular, and other representations.

Medical devices (such as defibrillators and insulin pumps) that are continuously linked with the patient for medical treatment are referred to as *active medical device(s)* (AMD). On the other hand, medical devices (such as home monitoring devices and medical beds) whose focus is on periodic monitoring of the patient physical condition and report generation are called *passive medical device(s)* (PMD). Wireless communication technologies are adopted for communication in IoT devices such as near-field communication (NFC), RFID, Wi-Fi, Bluetooth, LTE, and LoRA. Various IoMT devices use different wireless technologies. Most of the AMD and PMD utilize Bluetooth classic, V4.X, and V5. Bluetooth technology provides a generic profile for medical IoT devices to use the 2.4 GHz frequency band, as recommended by the international telecommunication unit (ITU) [17]. Some of the basic differences between BR/EDR and the BLE are showcased in Table 1.

Bluetooth-enabled devices have two modes of operation. In the single mode, a BLE device cannot interface with a device that is operating on BR/EDR, and vice versa. Whereas in dual-mode, both BR/EDR and BLE devices can communicate with each other. However, the major concern is about security and privacy in all Bluetooth versions. In this paper, we focus on the detection of attacks against the BR/EDR and BLE, since the medical sensor and data collection devices in the considered testbed utilize this version of Bluetooth.

### 2.3. Vulnerabilities in the Bluetooth Protocols

The major vulnerability factor in Bluetooth devices is the version that is used for communication. Table A1 in the Appendix A describes the vulnerabilities and security flaws of Bluetooth devices for different versions [18]. Few of the known vulnerabilities have been identified by researchers, such as MITM, Bluesmack, battery drain attacks, and backdoor attacks [19]. Recently, researchers identified the “SweynTooth” vulnerability affecting implantable medical devices (e.g., insulin pumps, pacemakers, and blood glucose monitors) and hospital equipment (e.g., patient monitors and ultrasound machines) that work on BLE [20]. The Bluetooth protocol has problems due to the encryption key length and improper storage of the link keys can be potentially manipulated by the adversary [12].

### 2.4. Intrusion Detection Systems

Some prior research studies on intrusion detection system(s) (IDS) dedicated to the cyber-physical system [21] or smart environments using the Wi-Fi protocol against DoS attack [21] have adopted various AI techniques, such as ML and DL. One such approach, Ref. [22], proposed a hybrid model that is based on the principal component analysis (PCA) and information gain (IG) incorporating the support vector machine (SVM), multi-layer perceptron (MLP), and instance-based learning models to identify the intrusions in the network. The model is trained and tested using the NSL-KDD, Kyoto 2006+, and ISCX 2012 datasets, and the optimal features are selected using an ensemble classifier. However, the performance of the model is evaluated with some publicly available datasets, which are not real-time datasets. Sawarna et al. [23] proposed an efficient IDS based on the deep neural network (DNN) using the principle component analysis–grey wolf optimization (PCA-GWO); it eliminates adversarial activities by providing faster alerts. This research was conducted to address the problem of data dimensionality for publicly available huge datasets. They tested the NSL-KDD dataset on various ML and DNN models to detect anomalies, among which the best accuracy was attained by the DNN. Baburaj et al. [24] proposed a cloud-based healthcare system using an SVM model to predict the health condition of a patient. The confidential data were accessed only by a legitimate user. This approach focused on data mining techniques using ML models, but not identify the anomalies in the system.

Likewise, a supervised approach for detecting intrusions in IoT devices in a smart home was proposed by Eanthi et al. [25]. In this approach, a lightweight standalone three-layer IDS framework is built using a decision tree (DT) classifier with promising results. Nevertheless, the evaluation of the proposed model is based on a simulation performed on the open-source Weka tool and the effectiveness of the IDS is not tested against real-time traffic and attacks.

### 2.5. IDS for Bluetooth Enabled Systems

Very few researchers have focused on the security perspective of Bluetooth technology, especially intrusion detection. Various attacks against Bluetooth devices are discussed below to emphasize the need for effective intrusion detection for Bluetooth-enabled medical IoT devices. Bluetooth technology provides a generic profile for the IoMT devices and it uses the 2.4 GHz frequency. It is identified as an attractive protocol for the healthcare system due to its robustness, lesser power consumption, low cost, suitability for short-distance communication, and support for data and audio streaming. Moreover, it helps in the IoT domain for machine-to-machine (M2M) communication [26]. Compromising the IoMT devices could lead to sensitive patient information being revealed through the interception and decoding of the data and audio/video streaming packets. An IDS detects malicious activities or policy violations that bypass the security mechanism on a network and is the process of monitoring and detecting unauthorized events intruding on the network. An intruder is one who escalates the privileges of the users to gain access to data or services or to control the entire network. Bluetooth-enabled systems require a different approach and standard IDS developed for other protocols are not effective due to the difference in traffic patterns and the highly constrained nature of Bluetooth devices [27].

Haataja et al. [28] proposed a Bluetooth intrusion detection and prevention system based on a set of rules by investigating Bluetooth security to discover malicious communication on the Bluetooth network. Krzysztoń et al. [29] proposed a detection system to identify the malicious behavior of Bluetooth traffic in a Bluetooth mesh network. Multiple watchdog nodes are used for cooperative decisions in different areas of the mesh network. Malicious activities are detected based on the received signal strength indicator (RSSI). However, this model encountered the problem of modeling the transmission range and RSSI parameters with obstacles, such as furniture and walls. This detection mechanism was not deployed to a variety of attacks and was evaluated in a simulated environment.

Similarly, Satam et al. [30] built a Bluetooth IDS (BIDS), where the normal behavior of the Bluetooth traffic was defined based on the n-gram approach, and malicious traffic was classified using traditional ML algorithms. This method attained the highest precision of about 99.6% and recall of 99.6% against DoS attacks. Yet, the effectiveness of the IDS was not tested against different datasets and other attacks. An anomaly-based intrusion detection system was proposed by Psatam et al. [31] to detect multiple attacks on the Bluetooth protocol using ML models by following the zero-trust principle. Nevertheless, the model was not tested using different attacks and datasets. Newaz et al. [32] focused on the detection of the BLE for multiple attacks using ML models to identify the abnormal behavior of the BLE traffic from the normal traffic pattern. The evaluation of the model was done on their own real-time traffic for an ideal dataset but was not tested on other datasets.

From the above literature and Table 2, it is observed that the existing IDS approaches that are dedicated to healthcare IoT systems are at the initial stage of development. Few of the proposed IDS have validated their models on the data of the network simulation (dataset) or on a small number of IoT devices, but they have not been tested on multiple datasets. Moreover, these proposed IDS models detect malicious activities on the network by identifying the traffic patterns as normal or abnormal. It is also important to identify the various types of attacks on the network. In the below subsection, we describe the healthcare system in use by this paper and the Bluetooth technology (BR/EDR and BLE) deployed.

## 3. Methodology

To understand the application of the proposed architecture, we consider a scenario of an IoMT system (i.e., smart healthcare system) that comprises multiple IoMT devices as shown in Figure 2. Vital information from the IoMT devices is transferred to edge devices and the cloud and is further sent to the medical staff.

### 3.1. Scalable Architecture

By considering the significant security mechanisms, we designed a scalable architecture to deliver appropriate patient details to the medical experts from patient care efficiently and without manipulation, i.e., tampering. Our ultimate goal is to provide a security mechanism to detect malicious activities against Bluetooth communication on the edge node. The proposed architecture has enforced security policies, and detection mechanisms at the edge cloud and edge nodes to ensure fast response and secure emergency services. Edge computing helps to process the data efficiently with a quicker response time and assists with the deployment of the IDS. Figure 2 represents the proposed architecture of smart healthcare for detecting malicious behaviors of ambulance-bound, Bluetooth-enabled IoT medical devices in the smart healthcare system.

As the complete information of the patient flows in and out through the medical IoT gateway, it allows for a potential attacking surface to compromise the complete system by (1) targeting the medical IoT gateway to manipulate information before sending it to the medical professional or by (2) launching DoS/DDoS or MITM attacks to make the information manipulated or unavailable. Such malevolent activities can potentially put the patient’s life at risk. To avoid such abrupt manipulation of the information, we enforced a multi-layer intrusion detection model on the edge nodes of the healthcare system. The detection system comprises two layers, namely, Layer_1 and Layer_2. Layer_1 is responsible for gathering patient information through a gateway and performing the preprocessing, feature engineering, and feature selection techniques using various ML algorithms. Layer_2 will detect the abnormal activities of the Bluetooth traffic on the edge node using a DNN classifier. Next, we describe in detail the features of each layer:

#### 3.1.1. Layer_1

Layer_1 receives data from various medical IoT devices. The data from IoT devices is received at medical IoT gateways to analyze and store on the edge node. The fetched information is deeply analyzed and processed before it is transmitted to the medical professional for diagnosis. On this layer, preprocessing, feature engineering, and feature selection techniques using various ML algorithms are performed. Data preprocessing helps to provide the privacy of the medical information from the IoT devices because the information received from IoT devices is in plain text that can be intercepted by adversaries to perform medium- and high-severity attacks [34]. Data preprocessing is performed to transform actual data into data compatible with ML/DL models. For this process, we used numericalization (where a string is converted into integer (stoi), and then encoded into tokenized sentences before feeding to any model) and normalization. Data preprocessing helps the model to be trained and tested quickly. It also increases the accuracy of classification. We provide a detailed explanation of these stages below.

*Eliminating/Dropping features*: While capturing the traffic, we eliminated some information, such as source and destination information, due to two major issues, firstly, in some scenarios, it is difficult for the sniffer to collect this information [33], while in other cases, the adversary may spoof its address giving wrong information. In both cases, the classifier attempted to misclassify the traffic by replacing the missing values with some random numbers, giving higher false positives and true negatives. Likewise, we eliminated some other unimportant and irrelevant features.

*Feature selection*: In this process, significant features were selected from the dataset by applying various feature selection techniques [35,36]. Feature selection increases the model performance, decreases computational cost, and also increases storage efficiency. Additionally, using appropriate features reduce the problem of overfitting.

There are various ML approaches for selecting features, such as filter-based methods, wrapper methods, embedded, and statistical methods. In the univariate selection technique, a statistical test is applied to each feature to select the features, which have a strong bond with the output variables. We used *Chi square (chi-2)*, in Equation (Equation 1), which gives the level of independence between the features x_t and the label y_t; it differentiates the chi-distribution, with the degree of freedom as 1.
(1)$$\chi^{2}\left(x_{t},y_{t}\right)=\frac{M\cdot {\left(FZ-PQ\right)}^{2}}{\left(F+P)(F+Q)(P+Z)(Q+Z\right)}$$
where *F* indicates the frequency of the features and their labels in a dataset; *P* = frequency of the features emerges without a label; *Q* = frequency of label emerges without features; *Z* = frequency of neither features nor label emerges in the given dataset; and *M* = no. of training samples $x_{t}$ = $x_{1},x_{2},\ldots x_{i}$ and prediction sequence $y_{t}$ = $y_{1},y_{2},\ldots y_{i}$.

*Recursive feature elimination (RFE)* is an effective method to find an optimal set of features for both regression and classification tasks. Initially, it creates a model dependent on all the features and estimates the importance of each feature of a given dataset. It priorities the features based on the rank order and eliminates those features that are of the least importance based on the evaluation metrics (in our case, we selected accuracy as a metric to find the optimal features) of the proposed model (DNN), which is depicted in Figure 3.

We also utilized logistic regression (LR) and random forest (RF) [37] to determine which features contributed to the output variable [38]. Table 3 and Table 4 show (“True” value), which indicates that the feature contributed to the output variable, based on each univariate selection algorithm. The final score is given based on the cumulative of the four algorithms used. In the BR/EDR and BLE dataset, they contain four and five non-numerical values, respectively. The non-numerical values are converted to numeric values before they are fed to the model using one-hot encoders, a process called numericalization. Finally, we only selected the features that were important for identifying abnormal activities.

*Normalization*: This is a feature engineering technique used to have the data in one range for faster processing and classifier accuracy. There are various normalization techniques available, among which Z-score normalization is highly used due to its simplicity and performance accuracy [33].

#### 3.1.2. Layer_2

Initially, the medical data from IoT devices is collected and pre-processed on the first layer, and the collected events from Layer_1 events are sent for detection and identification to the second layer (the edge node). If any manipulation or deviation in the Bluetooth traffic is identified, an alert is triggered. On this layer, the events of the IoT medical device are actively captured and recorded on the events collector and are placed on the EPCRD device. This traffic is fed in the format of a feature vector, which is represented in Equation (Equation 2).
(2)$$X\left(t\right)=\left(E_{1},E_{2},E_{3},\ldots ,E_{n}\right)$$

This feature vector is fed to Layer_2 to identify the malicious activities on this device based on the DL technique, which is deployed on the second layer of the edge node. The reason for placing two layers of intrusion detection is to protect the IoT system from device-based attacks and to have full coverage of the IoT healthcare network. The classifier model gives 99% accuracy, which has been placed on Layer_2. As the preprocessing and intrusion detection phases are separated on different devices, the resulting system constitutes a multi-layer IDS. At last, the IDS model triggers an alert for the administrator to take the required course of action against the intrusion.

### 3.2. Dataset Description

We developed a Bluetooth (BR/EDR and BLE) dataset using realistic traffic generated using the smart healthcare testbed [39] as described above in Figure 2, with the following specifications: GPU 128-core Maxwell, CPU Quad-core ARM A57 @1.43 GHz, and memory of 4 GB 64-bit LPDDR4 25.6 GB/s; this device is commonly known as NVIDIA Jetson Nano. The dataset comprises abstract meta-information from the network traffic flow link layer (data link) of the Bluetooth-enabled IoMT network. The generated data do not cover the exact patient vital information but we considered the payload size of the vital during data generation and transmission.

While generating the data, we used three IoMT devices that were easily available in the market (SpO2, heart rate, and ECG), which operated wirelessly. During the data generation process, we considered Bluetooth version.4 and above. We observed some delays in data transmission for DoS attacks. However, in a DDoS attack, the IoMT device stops sending the data transmissions, and the device malfunctions. The generated data are stored in the local drive of the edge node.

We collected 5 GB of BR/EDR and BLE data over about 76 h during normal traffic patterns and while performing the attacks. Therefore, the data collected included benign and malicious traffic. The performed attacks were DDoS, Bluesmack, MITM, and DoS on the L2CAP (link layer control adaption protocol) layer of the Bluetooth protocol stack. The L2CAP protocol was located in the data link layer of the stack, and it provided connectionless and connection-oriented data services to the top layer protocols. It allowed the upper-level protocols and applications to send and receive the data frames.

After analyzing the captured traffic in the preprocessing data, we used a Dell Precision T5820 workstation having the feature of Intel^®^ Xeon^®^ W-2245 (16.5 MB cache, 8 cores, 16 threads, 3.90 GHz to 4.70 GHz Turbo, 155 W), NVIDIA^®^ RTX™ A4000, 16 GB GDDR6, 4 DP. The data preparation process was done using Python libraries. These libraries are most efficient in the domain of data science (e.g., Pandas). Pandas supports various input and output data formats and has strong probabilities in estimating the statics and elementary visualization [40]. Finally, we selected nine features from each dataset through statistical methods and correlation analysis as presented in Table 3 and Table 4.

### 3.3. IDS Classifiers

The entire classification process is divided into two main stages—training and testing. In the training phase, some samples of a dataset are used to train the model. In the testing phase, new samples are fed to the classifier from the test dataset to evaluate the performance. To validate the dataset performance, we used existing supervised and unsupervised ML algorithms in addition to the proposed DL model for training and testing. The reason for using various ML and the proposed DL models is to benchmark it and to show that the dataset is free from abnormal results on different classifier models. Many of the datasets used in the literature are algorithm-dependent [41]. Our dataset produced acceptable accuracy for supervised and unsupervised ML and DL models. Various experiments with different classifiers helped us build the most efficient DL model to identify malicious activities with more than 99% accuracy.

#### 3.3.1. Classifier Using Supervised ML Algorithms

Among the existing supervised ML algorithms, we selected the most popular ones, namely: logistic regression (LR), decision tree (DT), support vector machine (SVM), and random forest (RF). We provide short descriptions of the algorithms that we used in experiments.

#### 3.3.2. Classifier Using Unsupervised ML Algorithms

The selected algorithms are naïve Bayes (NB), isolation forest (IF), K-Means (KM), and local outlier factor (LOF). Unsupervised algorithms are trained without using the labels of the features in the dataset. IoMT devices operate on different protocols, and due to this complexity, vulnerabilities may emerge. Furthermore, with classical ML algorithms, many attacks cannot be detected when the attacker does a small manipulation over time. DL techniques can recognize unknown patterns, outliers, and small changes from the training model.

#### 3.3.3. Classifier Using DNN

We used the multilayer perceptron (MLP) model, which is one of the categories of the feed-forward neural network (FNN), with multiple layers: one input layer, one output layer, and three hidden layers. Each layer consists of a set of neurons. The process of assembling the hidden layers is known as a DNN, as depicted in Figure 4. The DNN-IDS training comprises two phases—forward propagation and backward propagation. In forward propagation, output values are calculated. Whereas, in backward propagation, the weights are updated by passing the residual. The training of the model is implemented using Keras (with TensorFlow backend) and Table 5 provides detailed information on the various functions and parameters used. The combination of all layers is reflected in Figure 4. The model’s hidden layers are formulated as in the MLP. The vector and the biases are represented as $b_{h}$ and $b_{y}$.
(3)$$f\left(\theta \right)=L\left(y_{t}:{\hat{y}}_{t}\right)$$

Hidden layer:(4)$$Hl(x)=Hl_{1}\left(Hl_{1}-1\left(Hl-2\left(\ldots \left(Hl1\left(x\right)\right)\right)\right)\right)$$Training samples:(5)$$x_{t}=x_{1},x_{2},x_{3},x_{4},\ldots ,x_{i-1},x_{i}$$Hidden states:(6)$$h_{t}=h_{1},h_{2},h_{3},h_{4},\ldots ,h_{i-1},h_{i}$$Predictions of sequence:(7)$${\hat{y}}_{t}=y_{1},y_{2},y_{3},y_{4},\ldots y_{i-1},y_{i}$$Input-hidden weighted matrix:(8)$$Wl_{x}\cdot Wl_{h}$$Output-hidden weighted matrix:(9)$$Wl_{y}$$

The objective function of the model, defined as the single pair of the training example $\left(x_{t},y_{t}\right)$ is: **L** is described as the distance calculating the actual $y_{t}$ and ${\hat{y}}_{t}$ denote the prediction labels, η denotes the learning rate and *k* denotes the number of iterations. In DNN, each hidden layer uses a non-linear activation function to model the gradient error. Among various activation functions, *ReLU* gives faster performance and can train the model with a huge number of hidden layers. For maximizing the efficiency of the DNN, we built the model by considering the binary-cross entropy loss function, *ReLU* function, and *softmax* function with non-linear activation to achieve greater accuracy among the most substantial probability value of each class. In addition, we applied dropout techniques, to counter the problem of overfitting, by ignoring the randomly selected neurons. During this process, downstream neurons are ignored in the forward propagation and updated weights are not applied for the backward pass [42]. The neuron weights are settled within the network and are tuned for specific features. This effect on the network will result in less sensitivity to the definite weights of the neurons, which makes better generalization and is less likely to overfit the training data. In the below subsections, we show the experiments that we performed in the selection of IDS classifiers for the IDS models.

## 4. Experimental Results

To choose the best classifier for Intrusion detection, we trained and tested the BR/EDR and BLE Bluetooth datasets with supervised and unsupervised ML algorithms and DNN. The experimental results and discussion are provided below.

### 4.1. Unsupervised ML Algorithms

#### 4.1.1. BR/EDR Dataset

The BR/EDR dataset is trained and tested on four unsupervised ML algorithms with a balanced ratio of DOS attack and normal traffic pattern. We trained the four algorithms as binary classifiers to identify the DOS attack and normal traffic. The results achieved are shown in Table 6 and Figure 5. The naïve Bayes algorithm recorded the highest accuracy, precision, F1-score, and other favorable metrics among all the algorithms. The precision and recall scores of Isolation Forest achieved an acceptable level of prediction, while K-means and LOF achieve more than 55% and 30% of precision and recall, respectively. This suggests that these two algorithms are not suitable to train the IDS using the created BR/ EDR dataset. Moreover, the reason for lower precision and recall of LOF is a direct indication that the dataset is fully pre-processed. The dataset does not contain a high level of deviations and we performed intensive preprocessing on the dataset to make it normalized and free from outliers (in the Layer_1 of the IDS model). Furthermore, the features that have been selected are highly significant for the output class. The other three metrics are the F1 score, area under the ROC curve (AUC), and Cohen’s kappa scores. These metrics provide a homogeneous pattern to the previous three metrics for the Naïve Bayes classifier.

#### 4.1.2. BLE Dataset

Similarly, the BLE dataset was trained and tested on the same unsupervised algorithms, but we modeled those as multiclass classifiers to identify DoS, MITM, and normal traffic from the samples. The performances of the classifiers are shown in Figure 6. The numeric scores of each class are visible in Table 7. Among the four unsupervised algorithms, naïve Bayes records the highest accuracy scores of 98,78, and 80 for DoS, MITM, and normal traffic identification, respectively. Recall, precision, and other metrics fall close to the accuracy scores for the naïve Bayes classifier. Isolation forest, K-means, and LOF classifiers show better performances than the BR/EDR dataset with an average accuracy of 80% for three classes.

### 4.2. Supervised ML Algorithms

#### 4.2.1. BR/EDR Dataset

Likewise, the dataset BR/EDR was modeled as a binary classifier using four supervised ML algorithms each time, namely LR, DT, SVM, and RF to differentiate the DoS attack and normal traffic. The experimental results depicted in Figure 7 and Table 8 show that accuracy, precision, and recall are satisfactory for all classifiers. However, the RF classifier gave the highest score for all three metrics, followed by DT, SVM, and then LR. This is clear evidence that the classifier model and dataset are efficient in identifying malicious traffic of DoS attacks on Bluetooth medical IoT devices.

Figure 7 also records the F1-score, AUC score, and Cohen’s Kappa score, substantiating the inference that we deduced from the previous three metrics. Moreover, we can conclude that the dataset gives stable results using any of these supervised ML algorithms, of which RF and DT are the most recommended for general IoT devices and other networks. However, in the case of medical IoT devices, we need to choose a lightweight computationally inexpensive model. Among the tested algorithms, K-means (unsupervised) and SVM (supervised) are lightweight but they are computationally expensive in terms of training a model that is deployable on medical IoT devices. Nevertheless, the performance scores fall short for the real-time IDS model, so we investigated the DNN models using the created datasets.

#### 4.2.2. BLE Dataset

The results of the multi-class model trained using the BLE dataset with four different algorithms are shown in Figure 8 and Table 9. We observe that, unlike LR, the accuracy scores of the three supervised algorithms, DT, SVM, and RF lie between 95% and 98%. Though the average performance of the three algorithms, namely, DoS, MITM, and normal, is satisfactory, it is difficult to choose the best among these three. Moreover, neither one of the single classifiers give better performances for the three identification classes to suit the real-time IDS performance. LR records less than 50% accuracy and unstable scores for other metrics. Because of these shortcomings, we investigated the use of a DNN model for both of the datasets.

### 4.3. DNN Model

Two DNNs were modeled as binary and multi-class classifiers using BR/EDR and BLE datasets, respectively. The training accuracies of the two models were between 92% and 95%, as depicted in Figure 9. The testing accuracies were 98% and above for both models. From these results, we conclude that the classifier model using DNN was the best among all the other algorithms we tested. This deduction was bolstered by considering the training and testing loss scores in Figure 10. The training loss of the two models started at approximately 0.3 and then reached 0.15 as the learning process went on. Similarly, the lowest Test loss recorded was 0.01, which is an indication of a stable DNN model.

Additionally, to check the uniformity of the dataset, we tested various ratios of abnormal (malicious) and benign traffic patterns. The ratios of benign and abnormal patterns considered were 50–50, 75–25, and 80–20. Each time, the results that we achieved were consistent, which suggests that our dataset does not have any bias in the ratios of the traffic patterns. The accuracy scores of all the tests show that our dataset achieved less accuracy for unsupervised ML algorithms than for the supervised ML algorithms. From Table 10 and Figure 11, we deduce that the dataset can be considered a standard for training IDS models to identify DoS, DDoS, and Bluesmack attacks against Bluetooth IoMT devices. Moreover, in comparison to other models, our proposed model attained the best accuracy, as shown in Table 11.

## 5. Conclusions and Future Work

Bluetooth communication is widely adopted in IoMT devices due to its various benefits. Nevertheless, because of its simplicity as a personal wireless communication protocol, Bluetooth lacks security mechanisms, which may result in devastating outcomes for patients treated using wireless medical devices. As discussed, continuous monitoring of network activity is efficient in identifying cyber-attacks in most scenarios. We applied the same concept to Bluetooth-based medical IoT devices in a smart healthcare system. In this paper, we proposed a secure and scalable architecture and deployed the IDS on the edge nodes of the smart healthcare system. we explored the issues and limitations of Bluetooth communication technology in IoMT systems and current IDS for Bluetooth-enabled IoMT devices. The second outcome of this research is a standard Bluetooth dataset and a DNN-based classifier for Bluetooth traffic. To the best of our knowledge, this is the first intrusion detection dataset for the Bluetooth classic and BLE. From the results, we can see that the created dataset can be used to train the IDS model for identifying DoS, DDoS, and Bluesmack attacks on medical IoT devices operated using Bluetooth technology. We also deduce that the proposed IDS classifier using DNN gives more than 99% accuracy, precision, and recall, which outperforms the existing models for identifying Bluetooth-based attacks.

In the future, we plan to enhance the following critical areas of the proposed model. (1) We look forward to enlarging our dataset with more attack types, other than DoS, DDoS, and MITM. (2) We plan to include the attack data of other protocols, such as Wi-Fi. (3) We will aim to improve the intrusion detection classifier to identify those attacks efficiently on different datasets (by applying data fusion or feature fusion techniques). (4) Furthermore, we plan to develop a mitigation technique for the identified attacks from our model and to detect unknown attacks so that the architecture can be extended to include mitigation mechanisms for the identified attacks.

## Figures and Tables

**Figure 1 sensors-22-08280-f001:**
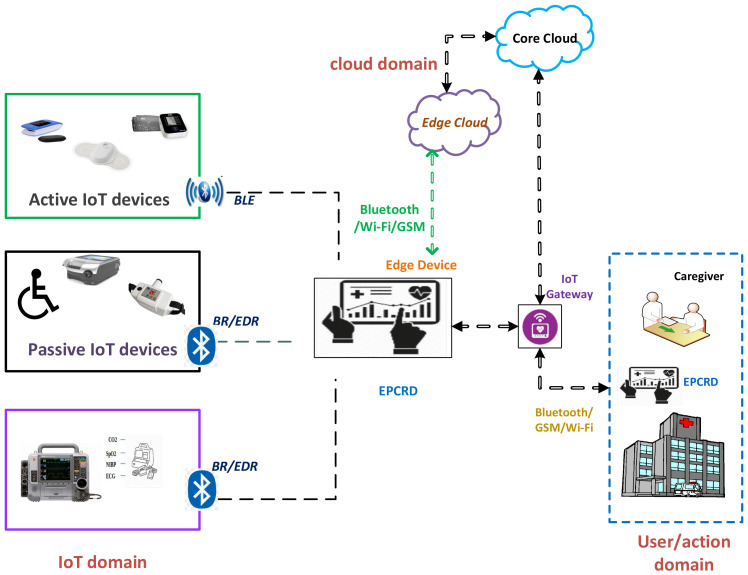
The use of Bluetooth and related protocols (BLE: Bluetooth low energy; BR/EDR: Bluetooth basic rate/enhanced data rate) in a typical smart healthcare system for communication between electronic patient care record device (EPCRD) and other entities over the edge and the cloud.

**Figure 2 sensors-22-08280-f002:**
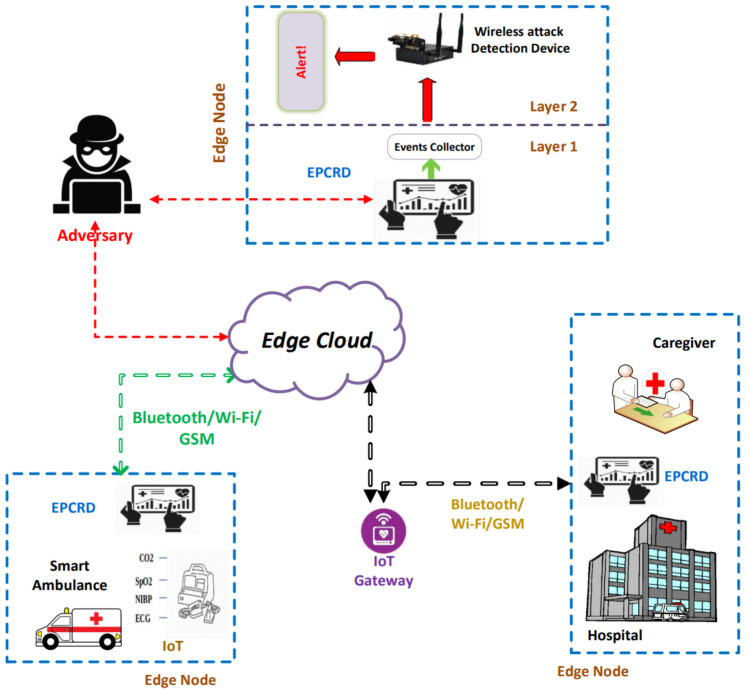
Architecture of the proposed security framework. The proposed system involves an edge cloud for reducing request/response delays. The IDS is multi-level and suits the resource restrictions of IoMT devices.

**Figure 3 sensors-22-08280-f003:**
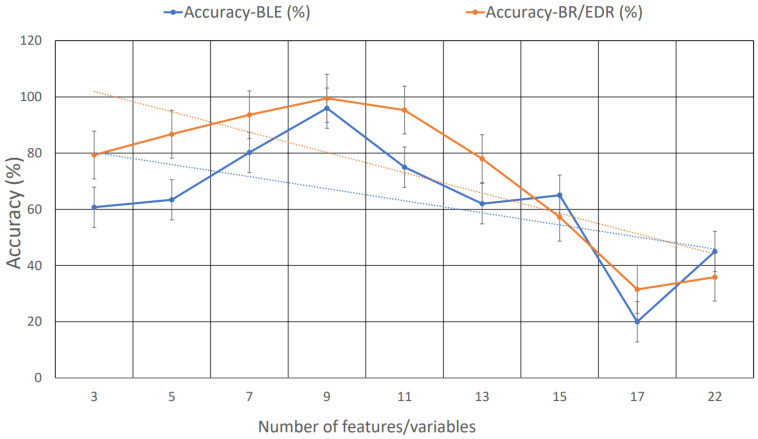
Accuracy of the model based on several features. Based on the varying accuracy of the number of features, we chose nine features from the dataset to train and test the model.

**Figure 4 sensors-22-08280-f004:**
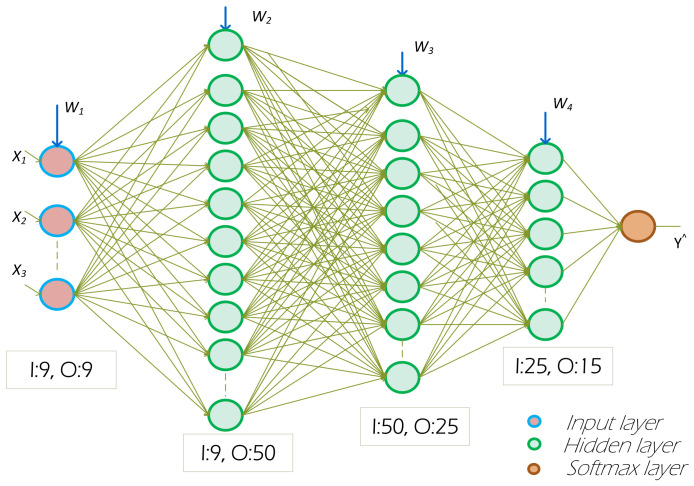
DNN architecture for the proposed IDS. It has three hidden layers with softmax as the output layer.

**Figure 5 sensors-22-08280-f005:**
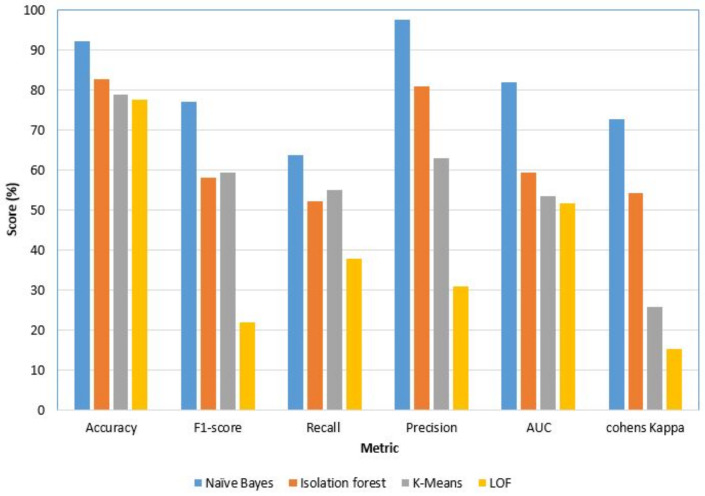
Performance of BR/EDR–Unsupervised ML algorithms. This result shows that the dataset does not show any deviation irrespective of different models (i.e., the dataset is preprocessed intensively).

**Figure 6 sensors-22-08280-f006:**
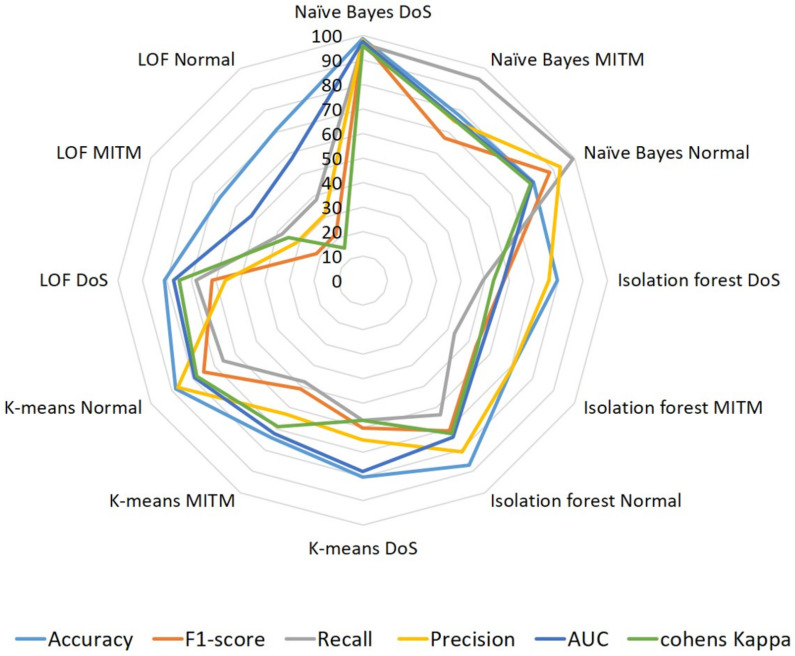
Performances of BLE–unsupervised ML algorithms. Multiple attacks were trained on the same models of BR/EDR; we observe that the models are not biased.

**Figure 7 sensors-22-08280-f007:**
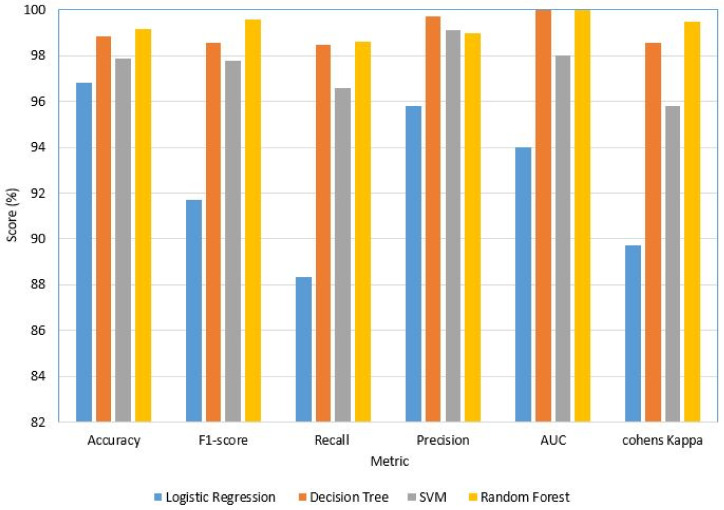
Performance of BR/EDR–supervised ML algorithms. The dataset and models are efficient in identifying malicious traffic behavior. (Deployed models are SVM and K-means).

**Figure 8 sensors-22-08280-f008:**
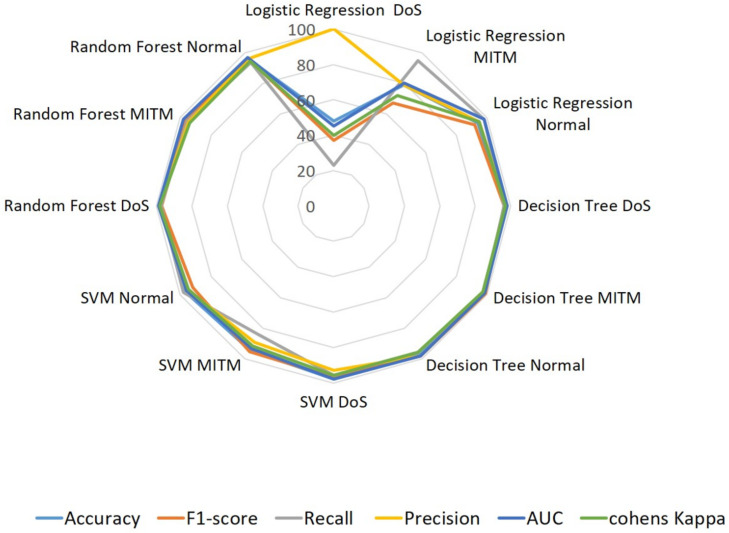
Performances of BLE–supervised ML algorithms. For real-time detection and deployment, neither of the single classifiers gave a better performance.

**Figure 9 sensors-22-08280-f009:**
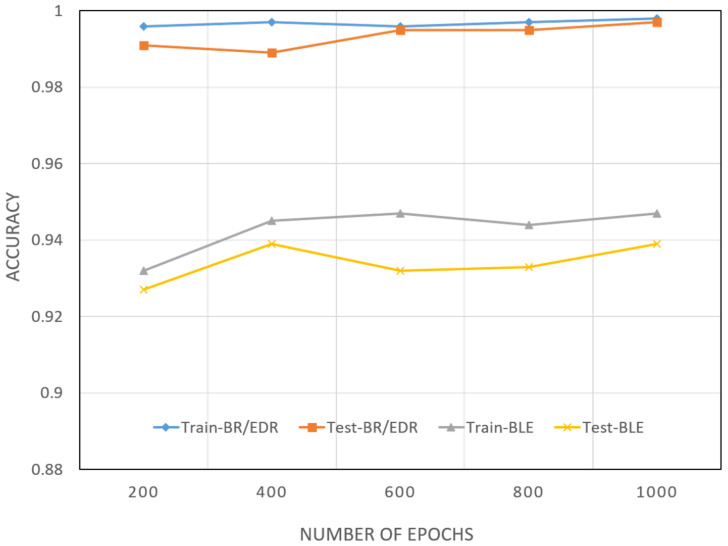
Training and testing accuracy. The proposed IDS DNN model for the BR/EDR and BLE datasets for 1000 epochs attained an accuracy of 98%.

**Figure 10 sensors-22-08280-f010:**
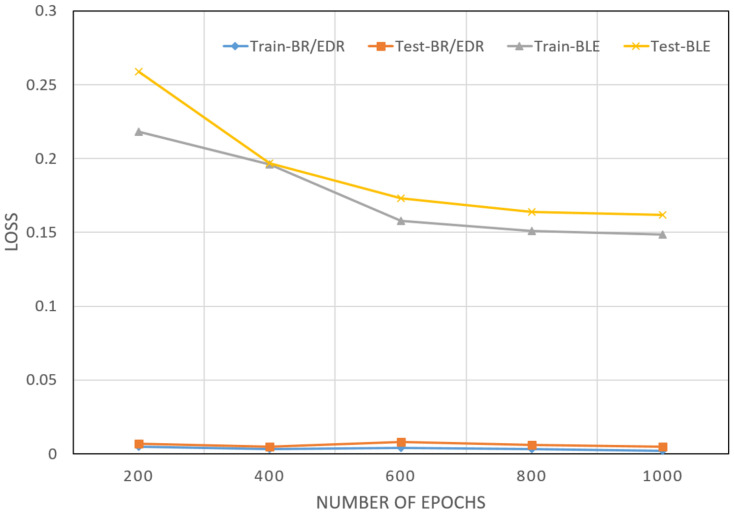
Training and testing loss–DNN. The recorded test was a loss of 0.01, which indicated that DNN was reliable for the real-time application.

**Figure 11 sensors-22-08280-f011:**
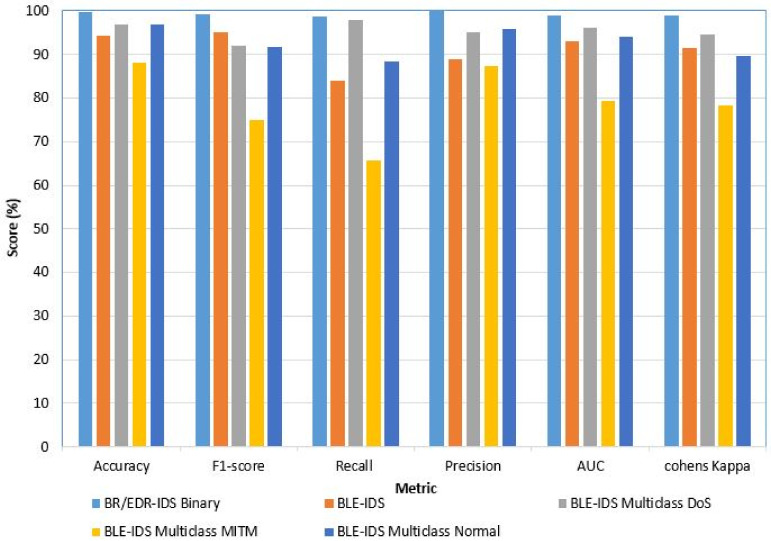
Performance analysis of the binary and multiclass of the proposed model for BR/EDR and BLE, respectively.

**Table 1 sensors-22-08280-t001:** Technical details of Bluetooth technology.

Features	Bluetooth Classic (BR/EDR)-2	Bluetooth V4.X (BLE)	Bluetooth V5 (BLE)
Medium access technique	Frequency hopping	Frequency hopping	Frequency hopping
Multihop solution	Yes	Yes	(Yes)
Network topology	Piconet, Scatternet	Star-bus mesh	Star-bus mesh
Radio frequency	2.4 GHz	2.4 GHz	(2.4 GHz)
Nominal data rate (Mb/s)	1–3	1	2
Distance range (m)	Up to 100	Up to 100	Up to 200
Latency(ms)	less than 100	less than 6	less than 3
Nodes/slaves	7	Unlimited	Unlimited
Message Size (bytes)	Up to 358	31	255

**Table 2 sensors-22-08280-t002:** Various BIDS approaches in comparison to our proposed models. Our Bluetooth intrusion detection covers both Bluetooth classic and Bluetooth low-energy protocols.

Article	Description	Model	Protocol	Data Used and Availability	Problem Address	Deployed
[23]	DNN-based IDS using (PCA-GWO)	Deep neural network (DNN)	Ethernet/Wi-Fi	(NSL-KDD)-Publicly available	Data dimensionality and anomalies detection	No
[24]	Cloud-based healthcare system	Support vector machine (SVM)	-	(Vital information)-No	Data mining techniques	No
[25]	Light-weight three-layer IDS for Smart home	Decision tree	Ethernet/Wi-Fi	(NSL-KDD)-publicly available	Simulation of anomaly detection	No
[28]	Bluetooth IDS for Bluetooth network	Defined set of rules	Bluetooth	(BR/EDR)-No	Malicious traffic detection	yes
[29]	Bluetooth mesh IDS-based on RSSI	Mesh network	Bluetooth	(BR/EDR RSSI signals)-No	simulation and detection of malicious patterns	Yes
[30]	BIDS for IoT	ML models	BR/EDR	(BR/EDR)-No	Malicious traffic based on n-gram	No
[31]	BIDS for IoT	ML models	BR/EDR	(BR/EDR)-No	Multiple attack detections based on zero-trust	No
[33]	BLE-IDS for medical devices	ML Models	BLE	(BLE)-No	Multiple attack detections for irregular traffic flow	Yes
Our approach	Bluetooth IDS for healthcare system	DL and ML models	BR/EDR and BLE	(BR/EDR, BLE)-yes	Multiple attack detection of BR/EDR, BLE traffic	Yes

**Table 3 sensors-22-08280-t003:** Univariate selection score of the BR/EDR selected feature.

Features	Chi-2	RFE	LR	RF	Score
btl2cap.length	True	True	True	True	4
HCI_events	True	True	True	True	4
HCI_ACL	True	True	True	True	4
Command Complete	True	True	True	True	4
Received direction	True	True	True	False	3
Sent Direction	True	True	False	True	3
Frame.cap_len	True	True	True	False	3
Disconnect complete	True	True	False	True	3
L2CAP	True	True	True	False	3

**Table 4 sensors-22-08280-t004:** Univariate selection score of BLE selected features.

Features	Chi-2	RFE	LR	RF	Score
btl2cap.length	True	True	True	True	4
Time	True	True	True	True	4
Protocol	True	True	True	True	4
Advertising_header_length	True	True	True	True	4
btle.access.address	True	True	True	True	4
PPI.DLT	True	True	True	False	3
btatt.opcode.method	True	True	False	True	3
btatt.opcode.command	True	True	False	True	3

**Table 5 sensors-22-08280-t005:** DNN architectural hyperparameters.

Description	Setting
Hidden Layer	3 (50, 25, 25)
Function	ReLU
Regularization	L2, dropout
Epochs	1000
Loss function	Binary_crossentropy
Optimizer	Adam
Batch Size	42
Dropout rate	0.025

**Table 6 sensors-22-08280-t006:** Performance analysis of the BR/EDR IDS using unsupervised—ML algorithms.

Metrics	Naïve Bayes (NB)	Isolation Forest (IR)	K-Means	LOF
Accuracy (%)	92.4	82.667	78.87	77.67
F1-score (%)	77.15	58.2	59.39	21.9
Recall (%)	63.68	52.34	55.01	38
Precision (%)	97.8	80.9	63.07	30.99
AUC (%)	82	59.38	53.48	51.62
Cohen’s Kappa (%)	72.86	54.34	25.87	15.2

**Table 7 sensors-22-08280-t007:** Performance analysis of the multiclass classification of the BLE IDS using supervised—ML algorithms.

	NB-DoS	NB-MITM	NB-Normal	IR-DoS	IR-MITM	IR-Normal	K-Means-DoS	K-Means-MITM	K-Means-Normal	LOF-DoS	LOF-MITM	LOF-Normal
Accuracy (%)	98.78	78	80.44	79.437	70.7	87.09	80.28	74.27	88.23	81	67.43	70.7
F1-score (%)	97.55	67	88	57.59	53.58	70.79	60.23	51.1	75	61.4	21.9	21.9
Recall(%)	96.78	95	99	49.2	43.12	63.31	57	47.71	65.78	68	38	38
Precision (%)	98.23	75	93	76.09	70.66	80.9	65	63.07	87.23	55.99	30.99	30.99
AUC (%)	97.55	76	80	57.34	55.687	73.93	77.87	72.13	79.43	77.12	52.62	57.62
Cohen’s Kappa (%)	96	75.34	79.32	53.56	53.98	72.34	57.23	69.06	78.21	75	35.2	15.2

**Table 8 sensors-22-08280-t008:** Performance analysis of the BR/EDR IDS using supervised–ML algorithms.

Metrics	LR	DT	SVM	RF
Accuracy (%)	96.8	98.85	97.89	99.15
F1-score (%)	91.7	98.59	97.8	99.6
Recall (%)	88.32	98.5	96.6	98.6
Precision (%)	95.8	99.7	99.1	99
AUC (%)	94	100	98	100
Cohen’s Kappa (%)	89.7	98.56	95.79	99.5

**Table 9 sensors-22-08280-t009:** Performance analysis of the multiclass classification of the BLE IDS using supervised–ML algorithms.

	LR-DoS	LR-MITM	LR-Normal	DT-DoS	DT-MITM	DT-Normal	SVM-DoS	SVM-MITM	SVM-Normal	RF-DoS	RF-MITM	RF-Normal
Accuracy (%)	48	79	94	96.63	98.5	97.29	97.89	94.39	96.86	97.74	96.5	95.78
F1-score (%)	37	67	92	96.27	99.12	97.8	96.8	95	92	97.27	96.12	95.66
Recall (%)	23	95	98	96.3	98.23	95.6	95.7	84	98	97.3	95.56	93.45
Precision (%)	100	79	95	97.5	98.43	98.1	93.1	89	95	98.5	94.7	96.23
AUC (%)	45	80	98	98	98.65	98	98	93	96	99	97.8	96.88
Cohen’s Kappa (%)	40	72	95	97	97.4	95.37	95.79	91.43	94.55	98	94	94.25

**Table 10 sensors-22-08280-t010:** Performance analysis of the binary and multi-class classification of the proposed IDS (BR/EDR and BLE).

	BR/EDR Binary-Class	BLE Binary-Class	BLE-DoS Multi-Class	BLE-MITM Multi-Class	BLE-Normal Multi-Class
Accuracy (%)	99.7	94.3	96.86	88.23	96.8
F1-score (%)	99.23	95	92	75	91.7
Recall (%)	98.65	84	98	65.78	88.32
Precision (%)	99.88	89	95	87.23	95.8
AUC (%)	99	93	96	79.43	94
Cohen’s Kappa (%)	99.08	91.43	94.55	78.21	89.7

**Table 11 sensors-22-08280-t011:** Comparison of our model with existing IDS models).

Model	Precision (%)	Recall (%)	F1 (%)	Accuracy (%)
[21]				
(Bluetooth)	98	98	97	98.4
[21]				
(Bluetooth)	96.7	88.23	91.8	97
[43]				
(Bluetooth)	88.64	88.64	87.5	-
[44]				
(NSL-KDD)	95.72	98.65	-	97.06
[45]				
(NSL-KDD)	96	98.7	97.3	-
[46]				
(NSL-KDD)	-	98.6	-	99
Proposed IDS				
(BR/EDR)	99.7	99.06	99.38	**99.8**
Proposed IDS				
(BLE)	95	98	95	**96.86**

## Data Availability

The BlueTack dataset is available at: IEEE Dataport under the title *BlueTack*, doi: https://dx.doi.org/10.21227/skhs-0b39.

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
