# Peer review of "Secure Bluetooth Communication in Smart Healthcare Systems: A Novel Community Dataset and Intrusion Detection System"

_sensors, 2022, doi:10.3390/s22218280_

Round 1

Reviewer 1 Report

In this paper, the authors propose an interesting and intelligent approach to building a DoS, DDoS and MITM secure smart health system for Bluetooth communicating objects. The authors used a set of models (supervised, unsupervised as well as DNN) for better intrusion detection. The results are promising with a rate of 99%. 

The paper is well written and also well structured. The contribution seems to me to be in three parts of which I will give my remarks for each part.

- Part 1: The architecture and framework implementation.

In figure 1, the authors have proposed an architecture based on BLE and BR/EDR. In this architecture, the devices (objects) communicate through Bluetooth to EPCRD to go to the Cloud (for which the authors do not give any visibility on the type of the platform) and or to the Gateway for a Bluetooth, Wi-Fi or GSM communication. In inpatient settings, the choice of communication protocols must be related to the health of the patients. An orientation towards Li-Fi seems to be a good approach. No specification for the use of the generated data nor the decision-making environment that usually is done by a doctor. The security of the data from the objects to the cloud is not taken into consideration. It seems to me that the authors should add identification keys (figure 2 just illustrates the access of an adversary as a foreign object).

- Part 2: Open database generation

The authors have created an open database combined with malicious traffic using DoS, DDoS, MIMTM. 9 features were selected with pre-processing already provided. Did the authors create the database in a real hospital environment? Or just Bluetooth based communications in closed spaces? Did the authors take into consideration the sending channel because it is physiological data and the data in this case is too sensitive.

-Part 3: Data processing and analysis

The authors used supervised, unsupervised and .net DNN methods for the MLP models. The choice of models is very wide which I appreciate. The authors talk about the accuracy which I prefer to have also semi-supervised models to be sure of the accuracy. It is necessary to integrate the confidence interval to consolidate the decision making. I have no idea about the pre-processing, the authors said they did that in collection which I am not sure about the results!

Reviewer 2 Report

The text is informative and written in proper English, it is pleasant to read. The novelty (BlueTack Dataset, etc.) is properly highlighted. The topic related with IoT and IoMT is interesting, important and up to date. Equations and mathematical formulas seem proper and free of error. However, there are some aspects that should be modified or corrected.

Suggestions and comments:

Check the Author’s Affiliation section and prepare it uniformly for all.

Several minor editorial and formatting issues are present, e.g., lack of space or multiple (unnecessary) space signs between subsequent words, blank lines or spacing between subsequent lines, etc.

Not all cited Author’s names are properly written, e.g., Krzysztoń (or Krzyszton) on page 6, etc.

Check the proper use of Capital letters in Table 2 and make necessary corrections.

Question about the dataset – how many devices did it include? Were they custom build or freely available on the market? What was the version of used Bluetooth standard? What was the distance and manner (wired or wireless) of data transmission? Were factors such as BER/BLER, delay/latency, monitored during operation? What other QoS aspects were monitored? How was it stored (on the local drive or build in memory of each and every device)? Additional comments are necessary.

Font style, type, size, etc., in case of all figures, should be uniform.

What kind of hardware (laboratory stand) and software (simulation environment, libraries, toolboxes) were utilized during the data processing? Additional comments are necessary.

In case of figures, all axes (both X and Y) should be properly labelled. What do they describe? What are those units?

The Conclusions and Future Works chapter should be extended. Do provide additional feedback and source of inspiration for other researchers and potential readers.

The number and quality of cited references is far too short. Authors need to look for additional papers focused on QoS and QoE aspects related with, e.g., mobile networks, cellular networks, transmission of packet data and multimedia, resource management, as well as surveys and user expectations studies, etc.

To sum up, the paper is good, however it requires a revision.

Round 2

Reviewer 2 Report

The Authors have addressed my suggestions and comments. This is a good and interesting paper. I do recommend it to be accepted and published.